# Anti-Obesity Effects of Polyphenol Intake: Current Status and Future Possibilities

**DOI:** 10.3390/ijms21165642

**Published:** 2020-08-06

**Authors:** Mariarosaria Boccellino, Stefania D’Angelo

**Affiliations:** 1Department of Precision Medicine, University of Campania “Luigi Vanvitelli”, 80138 Naples, Italy; mariarosaria.boccellino@unicampania.it; 2Department of Movement Sciences and Wellbeing, Parthenope University, 80133 Naples, Italy

**Keywords:** polyphenols, obesity, antioxidants, oxidative stress, inflammation, lifestyle

## Abstract

The prevalence of obesity has steadily increased worldwide over the past three decades. The conventional approaches to prevent or treat this syndrome and its associated complications include a balanced diet, an increase energy expenditure, and lifestyle modification. Multiple pharmacological and non-pharmacological interventions have been developed with the aim of improving obesity complications. Recently, the use of functional foods and their bioactive components is considered a new approach in the prevention and management of this disease. Due to their biological properties, polyphenols may be considered as nutraceuticals and food supplement recommended for different syndromes. Polyphenols are a class of naturally-occurring phytochemicals, some of which have been shown to modulate physiological and molecular pathways involved in energy metabolism. Polyphenols could act in the stimulation of β-oxidation, adipocyte differentiation inhibition, counteract oxidative stress, etc. In this narrative review, we considered the association between polyphenols (resveratrol, quercetin, curcumin, and some polyphenolic extracts) and obesity, focusing on human trials. The health effects of polyphenols depend on the amount consumed and their bioavailability. Some results are contrasting, probably due to the various study designs and lengths, variation among subjects (age, gender, ethnicity), and chemical forms of the dietary polyphenols used. But, in conclusion, the data so far obtained encourage the setting of new trials, necessary to validate benefic role of polyphenols in obese individuals.

## 1. Introduction

Obesity represents one of the main public health problems worldwide, both because its prevalence is constantly and worryingly increasing not only in western countries but also in low-medium income ones and because it is an important risk factor for various chronic diseases [1,2]. Obesity is when the body has too much fat, and this syndrome can cause a lot of damage to the body. It is very important to take steps to tackle obesity because, as well as causing obvious physical changes, it can lead to a number of serious and potentially life-threatening conditions. Obesity increases the risk of diseases, such as type 2 diabetes mellitus, fatty liver disease, hypertension, myocardial infarction, stroke, dementia, osteoarthritis, obstructive sleep apnea, and several cancers (such as breast, colon, and prostate), thereby contributing to a decline in both quality of life and life expectancy. Obesity can also lead to psychological problems, such as depression and low self-esteem [3].

It is estimated that 44% of cases of type 2 diabetes, 23% of cases of ischemic heart disease, and up to 41% of some cancers are attributable to obesity/overweight (Figure 1). In total, overweight and obesity represent the fifth most important risk factor for global mortality, and the deaths attributable to obesity are at least 2.8 million/year worldwide [4].

In particular, type 2 diabetes is the disease most associated with obesity; the term “diabesity” indicates that most diabetics are obese/overweight. By 2025, the prevalence of obesity-related diabetes is expected to double to 300 million [5]. If the trend is not reversed, by 2030, around 60% of the world’s population will be obese; therefore, according to the World Health Organization (WHO), obesity is currently a more severe world health problem than malnutrition [6]. Rising obesity rates and its association with other chronic diseases has dramatically increased healthcare costs [7]. 

During childhood and adolescence, alterations of body weight are associated with negative health consequences. Childhood obesity is progressively increasing worldwide, and an estimated 108 million children (under 20 years of age) suffer from obesity. Health conditions, such as type-2 diabetes and cardiovascular disease, usual only in adults have become increasingly common among children with obesity [8,9]. The food and beverages marketing in various media and contexts plays a fundamental role by significantly increasing the consumption of foods rich in sugar, high-fat, salts, and saturated fats by children and adolescents [10]. WHO has developed a series of recommendations, approved by the World Health Assembly, aimed at reducing the impact of the marketing of foods rich in saturated fat, trans-fatty acids, free sugars, or salt [11]. 

Obesity is a multifactorial problem which includes biological, behavioral, and social issues. Genetic factors can determine the number of fat cells or lead to alterations in eating behavior and energy expenditure. A sedentary lifestyle is another important factor that favors obesity; physical activity not only burns excess calories but also helps regulate food intake. Various endocrine factors can cause obesity: hyperinsulinism, hypercortisolism, ovarian dysfunction, and hypothyroidism. Finally, weight gain can be caused by taking medications, such as steroid hormones and psychoactive drugs [12]. A recent study has highlighted a close relationship between obesity and severity of SARS-CoV-2 disease, particularly in patients with a body mass index (BMI) ≥35 [13]. 

The evidence for social and environmental factors that contribute to obesity is often underappreciated. Obesity prevalence is significantly associated with sex, racial ethnic identity, and socioeconomic status, which creates complex relationships between each of these characteristics. Food availability remains an important factor associated with obesity that relates to differences in prevalence seen across geographical areas and higher rates of obesity within low socioeconomic status individuals [14]. Obesity prevalence differs by geographical region. Rural areas are associated with higher odds of obesity compared to urban areas. Rural areas tend to have farther distances between residences and supermarkets, clinical settings, and recreational opportunities, which may be impacting the ability to practice healthy behaviors that prevent obesity. This is one example of the “built environment”, which alludes to the infrastructure of a geographic area that influences proximity to and types of resources, transportation methods, and neighborhood quality. Depression, obesity in earlier age groups, short sleep duration, childhood abuse, and low maternal education have the strongest support among proposed risk factors for obesity [15]. Additionally, environments experiencing deprivation, disorder, or high crime have been shown to be associated with higher odds of obesity, which may appear more frequently in low social status individuals [14] (Figure 2). 

Proliferation of high calorie, energy dense food options that are or perceived as more affordable, combined with reductions in occupational and transportation related physical activity, can contribute to a sustained positive energy balance. The frequency and type of food vendors in a neighborhood determines the types of foods that residents can purchase. Historically, evidence has suggested that fast food restaurant density is associated with obesity prevalence [16] In addition to food availability and quality, the shift in food type, amount, and pricing is also relevant to the obesity epidemic. For example, available evidence strongly supports a greater risk of weight gain and type 2 diabetes with increased consumption of sugar-sweetened beverages [17].

The treatment of obesity is always inevitably associated with the reduction of body weight. This can be achieved through various strategies, which include lifestyle interventions (diet and exercise), normalizing the lipid metabolism, pharmaceutical interventions, or bariatric surgery [18]. Recent studies have shown that some natural dietary factors can influence body weight by representing a valid strategy to prevent obesity. In particular, the food intake of polyphenols can help reduce the physiological weight gain that occurs over time in the general population. 

Dietary interventions using natural bioactive food compounds have emerged as promising therapeutic tools for obesity and metabolic diseases, thanks to the limited side effects. The composition of the diet can affect metabolic and endocrine functions and the global energy balance [19]. Numerous studies show that bioactive compounds in the diet act as antioxidant and anti-inflammatory agents by increasing thermogenesis and energy expenditure and reducing oxidative stress, supporting progress towards weight loss and/or reduction of metabolic disorders [19,20].

This narrative review will summarize recent studies that document the potential role of both single polyphenol and some polyphenolic extracts in the modulation of obesity, focusing on human data.

## 2. Polyphenols: Bioactive Compounds

Polyphenols are a wide group of bioactive compounds naturally occurring in plants as secondary metabolites. They are a natural part of the human diet and evidence suggests that their consumption is associated with beneficial modulation of a number of health-related variables. It is generally accepted that the consumption of dietary polyphenols derived from fruit, vegetables, and other plant-derived foods may confer a number of health benefits [21,22,23,24,25].

There are currently about 8000 different polyphenols; these molecules share a common phenolic structure consisting of hydroxyl groups on an aromatic ring. Although polyphenols are chemically characterized as compounds with phenolic structural features, this group of natural products is highly diverse and contains several sub-groups of phenolic compounds. Fruits, vegetables, whole grains, and other types of foods and beverages, such as tea, chocolate, and wine, are rich sources of polyphenols [26]. The diversity and wide distribution of polyphenols in plants have led to different ways of categorizing these naturally occurring compounds. Polyphenols have been classified by their source of origin, biological function, and chemical structure. In addition, the majority of polyphenols in plants exist as glycosides with different sugar units and acylated sugars at different positions of the polyphenol skeletons [27].

Chemically, polyphenols are a large heterogeneous group of compounds characterized by hydroxylated phenyl moieties. Based on their chemical structure and complexity (i.e., the number of phenolic rings and substituting groups), polyphenols are generally classified into flavonoids and non-flavonoids (Figure 3).

Flavonoids form a major (over 9000 structurally distinct flavonoids have been identified in nature) heterogeneous subgroup comprising a variety of phenolic compounds with a common diphenyl-propane skeleton (C6-C3-C6). In turn, flavonoids are also classified into further subclasses according to their structural differences (flavanones, flavones, flavonols, flavan-3-ols or flavanols, anthocyanidins, isoflavones, and others, such as anthocyanins, proanthocyanidins, dihydro-flavonols) [26,28].

Flavonoids are built from a basic structure made up of an oxygenated heterocycle and 2 phenolic rings. They are distinguished by the oxidation state of the heterocyclic pyran ring, forming several groups (e.g., flavonols, flavanols, and anthocyanins). Until now, over 4000 flavonoids have been described in plants [27], and this number is constantly expanding due to multiple ways by which primary substituents are replaced, yielding more complex structures. Moreover, flavan-3-ols are also found in oligomer and polymer forms, known as proanthocyanidins. The nonflavonoids include phenolic acids (benzoic and hydroxycinnamic acids) and stilbenes. Further nonflavonoids that may be found in nature are gallotannins, ellagitannins, stilbene oligomers, and lignans [28,29]. Polyphenols present in fruit are principally quercetin and its glycosides; [+]-catechin, [–]-epicatechin, and their oligomers; procyanidins, anthocyanins, phloridzin, phloretin, and phenolic acids, e.g., chlorogenic acid [27].

Studies of the food diary suggest a wide variability in the consumption of polyphenols. In the United States, Spain, and Australia, it is estimated as a consumption of about 190, 313, and 454 mg/day of flavonoids, respectively [30]. The consumption of flavonoids in the diet is generally compensated by simpler phenols, as, for example, in Finland, where there is an average consumption of 222 mg/day of flavonoids and 640 mg/day of phenolic acids [31]. Likewise, a French cohort consumed an average of 1193 mg/day of phenols, of which phenolic acids make up 639 mg [32].

Polyphenols possess a wide range of beneficial effects against atherosclerosis, brain dysfunction, stroke, cardiovascular diseases, and cancer. Indeed, a dietary consumption of polyphenols was shown to be inversely associated with cardio- and cerebrovascular diseases due to the anti-inflammatory and antiatherogenic properties of polyphenols, such as inhibition of peroxyl radical-induced DNA strand breakage, inhibition of platelet aggregation and of the expression of adhesion molecules to the endothelium, and protection of low-density lipoprotein (LDL) from oxidative damage [33,34].

Evidence suggests that polyphenols can modulate/regulate or inhibit many cell signaling pathways in the human body. A large body of evidence speculates that polyphenols have protective effects against age-related degenerative diseases [35,36].

Although deficiencies in polyphenol intake do not result in specific deficiency diseases, adequate intake of polyphenols could confer health benefits, especially with regard to chronic diseases. Tea, cocoa, fruits, and berries, as well as vegetables, are rich in polyphenols. Flavan-3-ols from cocoa have been found to be associated with a reduced risk of stroke, myocardial infarction, and diabetes, as well as improvements in lipids, endothelial-dependent blood flow and blood pressure, insulin resistance, and systemic inflammation. The flavonoid quercetin and the stilbene resveratrol have also been associated with cardio metabolic health [33,34].

In recent decades, particular attention has been paid to the antioxidant or antiproliferative role of polyphenols [37,38,39,40,41,42] in the human diet, with evidence to support the contribution of polyphenols in the prevention of degenerative diseases [34,43]. Although oxidative stress has been observed during aging [44,45], under certain pathological conditions [46,47,48,49,50,51,52] or contractile activity [53,54], a number of studies have revealed that it is also related to the development of obesity. Excessive levels of reactive oxygen species (ROS) could lead to dysfunction of mitochondria by inhibiting the breathing process and resulting in reduced energy expenditure in adipocytes and, on the contrary, improvement of energy accumulation in adipose tissue.

Oxidative stress also suppresses the endocrine functions of adipose tissue by interrupting the secretion of adipokins, such as adiponectin [55]. Obesity is associated with low-grade chronic systemic inflammation in the adipose tissue, a condition influenced by the activation of the innate immune system in the adipose tissue that promotes the pro-inflammatory state and oxidation stress, triggering a systemic acute-phase response. Adipose tissue, especially in the visceral compartment, has been considered not only as a simple energy depository tissue but also as an active endocrine organ releasing a variety of biologically active molecules known as adipocytokines or adipokines. Based on the complex interplay between adipokines, obesity is also characterized by chronic low-grade inflammation with permanently increased oxidative stress [56]. Therefore, increasing the intake of antioxidants in the diet could have positive effects in obese patients. In addition, increased intake of polyphenols can therefore help to reduce body weight in elderly people at high cardiovascular risk [57].

## 3. Anti-Obesity Properties of Polyphenols

A number of in vitro studies have provided basic information for understanding the beneficial effects of polyphenols. In addition, many animal scientific research studies have shown that dietary supplementation with a polyphenolic extract is a potentially viable nutritional strategy for the prevention of obesity. In recent years, research has also focused on human experimentation, studying the action of single polyphenols and polyphenolic mixtures. In the next section, human studies and clinical trials using polyphenols to treat obesity are summarized.

### 3.1. Actions of Some Polyphenols

Curcumin (Figure 4) is the most bioactive polyphenol in the *Curcuma longa*, a plant usually consumed as a spice in India and other Asian states. It has been used for thousands of years in a medicine of Ayurveda, which means “science of long life”, and the first records about the turmeric as a useful medicine are dated 3000 B.C. Curcumin exerts several biological functions, including antioxidation, anti-inflammation, and antiangiogenesis, in different organs, including adipose tissue [58,59]. There is substantial evidence about effectiveness of curcumin in stimulating β-oxidation, inhibiting fatty acid synthesis, and decreasing fat storage [60,61].

A number of preclinical and clinical investigations have shown the beneficial effect of curcumin in attenuating body weight gain, improving insulin sensitivity, and preventing diabetes development [62,63,64].

Unlike the studies on the effects of curcumin in cells or animals, studies on obese subjects are limited. The first clinical trial using curcumin for obesity treatment was conducted by Mohammadi et al. [65]. In this study, obese subjects were treated with a commercial formulation of curcumin (1 g/day) supplemented with a bioavailability enhancer, piperine, for a month. Although there were no changes in weight, BMI, or body fat, serum triglyceride levels were significantly decreased after curcumin treatment, indicating the improvement of insulin actions [65] (Table 1). In another randomized study, Ganjali and Sahebkar showed that 30-day treatment of curcumin (500 mg/day) reduced serum levels of inflammatory cytokines IL-1β and IL-4 of obese individuals [66], indicating the anti-inflammatory activity of curcumin in obesity therapy. Moreover, oral curcumin supplementation (1 g/day for 30 days) was effective in reducing oxidative stress burden in obese individuals [67]. A randomized control trial conducted by Di Pierro et al. [68], among overweight people affected by metabolic syndrome, showed the ability of curcumin to reduce weight and omental adipose tissue. Curcumin supplementation had beneficial effects on body mass index, waist circumference, hip circumference, high-density lipoprotein levels, and triglyceride/high-density lipoprotein ratio in overweight and obese female adolescents [69] (Table 1).

Overall, these research studies show that curcumin applies anti-obesity and anti-inflammatory actions in part through adipose tissue, by decreasing adiposity, lipid storage, and enhancing lipid oxidation [70]. Although curcumin has been used in clinical trials, its multifaceted pharmacological nature, its pharmacokinetics, and its side effects in obesity therapy need to be carefully investigated. The recommended maximum daily usage of curcumin is 1 mg/kg body weight by a joint report of the World Health Organization (WHO) and the Food and Agriculture Organization [71]. Moreover, a few studies showed that the chronic use of curcumin can cause some upset [72]. Recent studies suggest that the metabolic effects of curcumin are linked to changes in the gut microbiota. However, research into curcumin continues to provide novel insights into metabolic regulation that may ultimately translate into effective therapy [73].

Quercetin (Figure 5) is one of the major flavonoids, and one of the most potent antioxidants of plant origin. It is found in many foods, including vegetables, such as onions, garlic, and ginger, fruit, such as apples, and wine. Although many in vitro and in vivo studies focused on the beneficial effects of quercetin in obesity, there are only a limited number of human studies and clinical trials that have been performed to evaluate the effects of quercetin on obesity treatment [70,74,75]. A study evaluated the effects of taking quercetin in overweight obese subjects with various apolipoprotein E genotypes; the quercetin (150 mg/day/subject) decreased the waist circumference and triacylglycerol concentration [76]. Quercetin supplementation improved some risk factors of cardiovascular disease, yet exerted slightly pro-inflammatory effects [76]. One hundred and sixty-two milligrams per day quercetin supplementation lowers ambulatory blood pressure in overweight-to-obese patients, suggesting a cardio protective effect of this polyphenol [77] (Table 1).

Although quercetin suppressed oxidative stress in obese animal models, Shanely et al. reported that quercetin has no effect on oxidative stress and antioxidant (500 or 1000 mg/day/subject for 12 weeks) in obese subjects [78].

Currently, a clinical trial that is still under phase II stage investigation, is investigating whether quercetin changes the absorption of glucose by the body in obese subjects [79]. Future research needs to further investigate the bioactive effects and bioavailability of quercetin.

Resveratrol (RE) (Figure 6) is a polyphenolic phytoalexin found in over 70 plant species and is highly concentrated in the skin of red grapes. Tea, berries, pomegranates, nuts, blueberries, and dark chocolate contain this polyphenol at varying concentrations [80]. RE exhibit a plethora of therapeutic benefits, including anti-inflammatory and antioxidant [81]. RE was discovered to be an activator of sirtuin 1, an important molecular target regulating cellular energy metabolism and mitochondrial homeostasis. An important target of RE is adenosine monophosphate-activated protein kinase (AMPK), suggesting that it can play a role in regulating energy homeostasis; by activating AMPK, RE exerts a lipid-lowering effect. RE has potential anti-obesity effects by inhibiting differentiation and decreasing proliferation of adipocytes, decreasing lipogenesis, and promoting lipolysis and β-oxidation [82].

While the effects of RE have been widely studied in animal models [83,84], few clinical studies have been performed, and the results are inconclusive. Moreover, contrary to the substantial preclinical findings of beneficial metabolic effects of RE in an obesity or inflammation setting, the outcome of human clinical trials of resveratrol effects on obesity-related morbidities have been inconsistent. In a cross-over study, Timmers et al. showed that 150 mg/day of RE treatment increased energy expenditure, reduced serum inflammatory markers, and decreased adipose tissue lipolysis and plasma fatty acid and glycerol levels of obese men [85]. In another study, Konings et al. investigated the effects of 30 days RE treatment (150 mg/day) on the adipocyte size and gene expression patterns in obese men. The authors found that RE treatment decreased the size of abdominal subcutaneous adipocytes [86] (Table 1). Some beneficial effects have also been observed in some clinical trials, although many discrepancies and conflicting information exist [87]. Arzola-Paniagua et al. [88] observed not significant decreases in BMI, waist circumference, fat mass, triglycerides, leptin, and leptin/adiponectin ratio in obese individual’s treated with resveratrol therapy. RE treatment did not improve inflammatory status, glucose homeostasis, blood pressure, nor hepatic lipid content in middle-aged men with metabolic syndrome. On the contrary, it significantly increased total cholesterol and LDL cholesterol [89]. RE is used combined with other phytochemicals, too. For example, twelve weeks of combined epigallocatechin-3-gallate and RE supplementation increased mitochondrial capacity and stimulated fat oxidation in obese humans [90].

However, another report showed that high levels of RE supplementation treatment had no effect on energy expenditure, adipose tissue content, and metabolic events. A clinical trial by Poulsen et al. did not seem to support the anti-obesity potential of RE in obese men; no effect was seen on blood pressure, resting energy expenditure, oxidation rates of lipid, ectopic or visceral fat content, or inflammatory and metabolic biomarkers [91]. It has been observed that RE presents a dose–response hormesis in the biological models in which it has been tested, affecting several outcomes with medical and therapeutic significance [92].

Evidence from these limited clinical studies combined with the results from in vitro and animal studies indicate that potential anti-obesity effects of RE may be achieved through dietary supplementation. The differing results could be due to the variable doses selected in the assays and to the different clinical backgrounds of the study subjects. However, the optimal doses and study period for the anti-obesity potential of RE remain to be determined [70,89].

### 3.2. Actions of Some Polyphenolic-Food

More numerous are the clinical studies in which the anti-obesity effect of extracts of polyphenolic foods is analyzed.

Many foods, such as apples, blueberries, gooseberries, grape seeds, kiwi, strawberries, green tea, red wine, beer, cacao liquor, chocolate, and cocoa, are rich in polyphenols. The effects of green tea extract (GTE) on obesity have received increasing attention. GTE is rich in polyphenols, including epigallocatechin-gallate (EGCG), epicatechin, epigallocatechin, and epicatechin-gallate: a 2-g bag of green tea contains about 500 mg of catechins [93]. Since the 1990s, green tea is seen as a natural herb that can enhance energy expenditure and fat oxidation, thereby inducing weight loss. In a double-blind study, 46 obese patients received either 379 mg of green tea extract. Three months of GTE supplementation resulted in decreases in body mass index, waist circumference, levels of total cholesterol, low-density cholesterol, and triglyceride. Increases in total antioxidant levels and serum concentrations of zinc and magnesium were also observed. These findings demonstrated that green tea could influence the body’s mineral status, and they showed the beneficial effects of green tea extract supplementation on body mass index, lipid profile, and total antioxidant status in patients with obesity [94]. In another study, 12 weeks of treatment with high-dose GTE resulted in significant weight loss, reduced waist circumference, and a consistent decrease in total cholesterol and LDL plasma levels without any side effects or adverse effects in women with obesity. The antiobesity mechanism of high-dose green tea extract might be associated in part with ghrelin secretion inhibition, leading to increased adiponectin levels [95,96]. Results of randomized, controlled intervention trials have shown that consumption of GTE (270 mg to 1200 mg/day) may reduce body weight and fat [97]. Almost all of the studies conducted with Asian subjects have shown positive results about the anti-obesity effects of tea extract; on the other hand, studies with Caucasian subjects reported mixed results [98]. There are several proposed mechanisms whereby GTE may influence body weight and composition. Green tea also contains caffeine, and there is probably a synergistic action between the different molecules. The fact that an EGCG–caffeine mixture stimulates energy expenditure cannot be completely attributed to its caffeine content because the thermogenic effect of an EGCG–caffeine mixture is greater than that of an equivalent amount of caffeine [99]. The predominating hypothesis is that GTE influences sympathetic nervous system activity, increasing energy expenditure and promoting the oxidation of fat. Other potential mechanisms include modifications in appetite, up-regulation of enzymes involved in hepatic fat oxidation, and decreased nutrient absorption [97].

In conclusion, cellular and animal studies have shown that dietary supplementation with green tea extract is a potentially viable nutritional strategy for the prevention of obesity. However, the efficacy of green tea remains unclear [100]. The low bioavailability of GTE along with potential confounders may have contributed to the inconsistent outcome of human studies [101]. Human trials of course have greater reproducibility difficulties. The discrepancies among the studies employing green tea may be due to the varieties of study designs, the length of study, age and gender of subjects, the ethnicity of subjects, the formulations of green tea supplement, and the presence or absence of weight control factors (i.e., caffeine, exercise, low-caloric diet). While few epidemiological and clinical studies show the health benefits of EGCG on obesity, the mechanisms of its actions are emerging based on the various laboratory data. These mechanisms may be related to certain pathways, such as through the modulations of energy balance, endocrine systems, food intake, lipid and carbohydrate metabolism, the redox status, and activities of different types of cells (i.e., fat, liver, muscle, and beta-pancreatic cells) [102].

Citrus fruits and their juices are natural sources of many bioactive compounds, such as flavonoids and carotenoids, and their properties, such as antioxidant power, derives from these molecules, too [103]. Particularly, grapefruit and bergamot are rich in citrus flavonoids, including naringenin, hesperitin, nobiletin, and tangeretin, which have emerged as potential therapeutics for the treatment of metabolic deregulation [104]. In obese humans, 1/2 grapefruit (49 mg naringenin) three times daily for 8 to 12 weeks reduced body weight and waist circumference [105]. Improvements were observed in circulating lipids, with total cholesterol and low-density lipoprotein significantly decreasing [105]. Grapefruit should be further evaluated in the context of obesity and cardiovascular disease prevention.

Orange juice consumption can promote lower levels of oxidative stress and inflammation due to the antioxidant activity of citrus flavonoids, too. Red orange juice was effective in reducing metabolic markers and inflammatory markers in human subjects [106], but it does not inhibit weight loss. It ameliorated the insulin sensitivity, lipid profile [107], and inflammatory status, as well as contributes nutritionally to the quality of the diet [108]. The authors suggest the consumption of freshly squeezed orange juice, without added sugars, as part of a controlled diet in obese women; the combination with increasing physical activity for maintaining a healthy body weight through healthier food choices could improve the comorbidities related to the obesity [107].

It is opportune encouraging the consumption of whole fruits and replacing packaged fruit juices with fresh fruit juices or plain water, as part of a broader set of dietary strategies to reduce total dietary energy intake in adult populations.

Several studies have demonstrated that mulberry, blueberry, cherry, black elderberry, blackcurrant, cherry, red cabbage, strawberry, black rice, purple corn, and sweet potato have beneficial effects on body weight management in various experimental models. These fruits are rich in anthocyanins, polyphenolic compounds commonly found in the daily diet, in particular of pigmented fruits, with potential pharmacological activities, including anti-inflammatory effects, antioxidants, and antiobesity [109,110,111]. Mulberries (*Morus alba* L.) have long been used as conventional medicine in Asian countries, and they have various pharmacological effects, such as hypolipidemia, antidiabetic, antioxidant, and anti-inflammatory effects, thanks to the presence of flavonoids and anthocyanins [112,113,114]. Mulberry fruit extract may suppress adipogenesis through modulating miR-21/143 expression and AMPK activity in 3T3-L1 adipocytes [115]. The extract of Japanese white mulberry blocks the alpha-glucosidase and then the intestinal hydrolysis of polysaccharides, thereby reducing the glycemic index of carbohydrates. Forty-six overweight people received 2400 mg of white Japanese mulberry extract. Weight loss has been about 9 kg in 3 months, equal to approximately 10 percent of the initial weight; the plasma insulin and glucose curves at the end of the trial were lower than those performed at the time of enrollment. Waist circumference and thigh circumference decreased [116]. The extract of mulberry may represent a reliable adjuvant therapy in the dietetic treatment of some patients who are obese or overweight [116]. Supplementation with bilberry juice resulted in significant decreases in plasma concentrations of C-reactive protein, IL-6, and IL-15. These data suggest that supplementation with bilberry polyphenols may modulate the inflammation processes [117]. In clinical studies, strawberry extract, rich in anthocyanins, pelargonidin sulfate, and pelargonidin-3-*O*-glucoside reduced the postprandial inflammation, plasma IL-6, as well as increased insulin sensitivity [118] and reduced oxidized low-density lipoprotein [119], in overweight human subjects.

The apple is rich in polyphenols, such as hydroxycinnamic acids, flavonols, dihydrochalcones, and anthocyanidins; the 22% of the dietary phenolic ingested comes from the apples. Daily consumption of apple for 4 weeks increased antioxidant enzymes in an elderly population, likely because of the presence of flavonoids [120]. Apples are associated with decreased risk of obesity in children according to the National Health and Nutrition Examination Survey (2003–2010) [121]. Dried apple can be recommended as a snack for overweight and obese children thanks to its high-fiber and polyphenol content [122]. Thirty-eight overweight or obese children consumed 120 kcal serving per day of either dried apple or a control snack (muffin) for 8 weeks; high-density lipoprotein cholesterol concentration increased within the apple group. Some intervention clinical trials with an apple juice showed a significant reduction in body fat mass but not in body weight, BMI, nor waist circumference [123]. Future research should evaluate the effects of consuming fresh apples that include the peel [122].

The **onion** is rich in quercetin. A study showed that 12-week of onion extract intake decreased body weight, percentage of body fat, and BMI of 10 female students [124]. The consumption of onion peel extract improves endothelial function in healthy overweight and obese individuals [77,125]. Moreover, Lee et al. demonstrated that onion (100 mg/day/subject) significantly decreased the total body fat, particularly in the percentage of fat in the arm, and decreased the BMI of overweight or obese subjects [126].

**Soybean** (*Glycine max*), an East Asian legume, has been approved as a health-promoting food because of its effects in prevention of metabolic disorders, such as hyperlipidemia, cardiovascular diseases, and type 2 diabetes [127,128]. These beneficial effects are supposed to be exerted by constituents, such as unsaturated fats, fiber, the high content of protein, and isoflavones [129]. Isoflavones, i.e., genistein, daidzen, and glycetin, have comparable chemical structures to endogenous estrogens and are believed to interact with intracellular estrogen receptors, which results a possible action in decreases in lipids accumulation and adipose distribution. Both animal and human studies have shown the effect of dietary soy on weight control and prevention of obesity [127]. Studies have revealed that isoflavones are involved in the inhibition of adipogenesis and lipogenesis by interrelating with several transcription factors and upstream signaling molecules [130]. Although the biological mechanisms that cause the actions of soy isoflavones and numerous effects remain unknown, it is noteworthy that isoflavones exhibit pleiotropic properties in the human body to control metabolism and balance, which may potentially inhibit and treat obesity [130]. Various studies show a role of soy in improving health; it causes a decrease in cholesterol, triglycerides, and adiposity, it acts on improving inflammatory response and defense against cancer, osteoporosis, and menopause. Positive soybean actions on inflammation have been reported; soy extract constrain secretion of inflammatory cytokines (TNF-α, MCP-1, and IL-6) and decrease the inflammatory responses in adipocytes [131]. Eighty-seven obese postmenopausal women took soy extract daily, corresponded to 80 mg of isoflavones. After 6 months, serum leptin and TNF-alpha levels declined, and a significant increase of serum levels of adiponectin was detected [132]. Soy-isoflavones studies showed that this extract may reduce body mass index in women, especially in dosages <100 mg/d and in intervention periods of 2–6 mo. In addition,, a trend for reduced BMI after their consumption was observed in Caucasians. Overall, results showed that soy-isoflavones extract may have different impacts on weight status [133]. Overweight human subjects were treated with black soybean extract. This randomized clinical study showed reduction in abdominal fat, cholesterol, triacylglycerol, and LDL levels, with decreasing TNF-α and MCP-1 levels [134]. Studies in obesity models have demonstrated that dietary soy-isoflavones significantly reduced body weight and fat pad weight. It was shown that sterol regulatory element-binding protein 1 SREBP1 transcription factor expression and its other target genes (acetyl-coenzyme A carboxylase, ATP-citrate lyase, and FASN) were reduced by soy isoflavones. Overall, available studies prove that soy-isoflavones decrease adiposity and inflammation, by decreasing lipogenesis, adipogenesis, and improving lipid clearance; they are capable of decreasing oxidative stress and may contribute in metabolic enhancements in obesity.

The search for polyphenolic extracts with antiobesity activity is always active, always new phytochemicals are proposed. *Artemisia princeps*, the Korean mugwort, is a common edible plant native to Korea, Japan, and China. In Western and African folk medicine, several species of the genus *Artemisia* are used for their claimed healing properties and for the cure of specific ailments. The early in vitro antiobesity-effects [135] of Artemisia could be due to the presence of active compounds, such as terpenes, steroids, and saponins, and also the active ingredients, such as tannins and flavonoids [136]. *Physalis alkekengi*, also referred to as ground cherry, is an indigenous herb in Iran and many other regions of Asia such as China. Preliminaries studies have reported presence of several active compounds, including secosteroids called physalins, and flavonoids [137,138]. Single compounds found in *Physalis alkekengi* should be studied separately to identify the main compound exerting in vivo antiobesity properties [139]. Persimmon (*Diospyros kaki*) fruit is native to China, and its cultivation extends to other part of East Asia, including Japan, where it is very popular, and it has high proanthocyanidin-type content [140]. Persimmon fruits possess high levels of dietary fiber, vitamin C, catechin, and gallocatechin [141]; they exert beneficial effects on obesity by inhibiting adipogenesis and reducing lipid synthesis and accumulation by regulation of lipid-related transcription factors in the white adipose tissue of obese mice [140].

The authors of the papers, whose data are reported in Table 2, support that the antiobesity effects observed can be attributed to the polyphenols present in the extracts indicated. Different discourse is instead with regard to the most recent phytochemical extracts (such as those obtained from *Physalis alkekengiad*, for example); for these new phytochemicals extracts, the samples should be further analyzed to determine which of the phytochemicals present in the extracts are responsible for the observed anti-obesity effects.

## 4. Bioavailability and Pharmacokinetics

Bioavailability is usually defined as the fraction of an ingested nutrient or compound that reaches the systemic circulation and the specific sites where it can exert its biological activity. To establish conclusive evidence for the effectiveness of dietary polyphenols in obesity and human health improvement, it is useful to better define the bioavailability of polyphenols [142]. The health effects of polyphenols in human depend on their absorption, distribution, metabolism, and elimination. The chemical structure of polyphenols determines their rate and extent of absorption, as well as the nature of the metabolites present in the plasma and tissues. Polyphenols are present in plants as aglycones, glycosides, esters, or polymers. Aglycones can be absorbed from the small intestine, while glycosides, esters, and polymers must be hydrolyzed by intestinal enzymes, or by the colonic microflora, before they can be absorbed. Certain chemical characteristics, for example, molecular weight, lipophilicity, stereochemistry, and the presence of a group capable of hydrogen bonding, affect the transport and permeability of the polyphenols into the cytosol enterocytes from the gut lumen [143]. Through the gastro-intestinal tract, gallic acid and isoflavones which has small molecular weight are easily absorbed [144]. On the other hand, numerous phenolic compounds absorbed a rate of 0.3–43%, and the metabolite content circulating in the plasma can be low. Hence, numerous factors affected may impart interference in the direct bioavailability of the phenolic compounds present in the food [145].

Catechins are unstable under physiologic conditions, and they could be rapidly degraded or metabolized through interactions with the hydroxyl groups on the phenol rings. Only a small fraction of catechins present in the intestinal tract after drinking tea can be absorbed, and therefore be considered bioavailable, i.e., appearing in the blood and tissues or reaching the systemic circulation. A study demonstrates that approximately 1.68% of ingested catechins were present in human’s plasma (0.16%), urine (1.1%), and feces (0.42%) after tea ingestion over 6 h [145]. A fraction of the ingested catechins undergoes extensive metabolism by phase II enzymes, such as UDP-glucuronosyltransferases, sulphotransferases, and catechol-*O*-methyltransferase, before and after being absorbed predominantly in the small intestine and in liver [32], with the remained catechins entering the colon. The metabolites identified in humans include glucuronide and sulfate conjugates, methylated catechin conjugates, and microflora-mediated ring fission products and phenolic acid catabolites [146]. Therefore, the therapeutic effect is limited [147].

The commercial use of RE as a pharmaceutical drug is currently facing several limitations because of its low bioavailability and rapid metabolism [148]. After consumption, more than 70% of RE is absorbed by the gastrointestinal tract, but it is later metabolized by three distinct metabolic pathways leading to its very low bioavailability. The RE is metabolized in the liver to glucuronide and sulphate forms and in the intestine by hydrogenation of the aliphatic double bond. The bioavailability and pharmacokinetics of this phenol depend on the doses ingested, the concomitant ingestion of food matrix, the particle size, the gut microbiota, and the circadian variation [149].

Only a minor proportion of quercetin is absorbed in the stomach: the primary site of absorption is the small intestine. The absorbed “unit” of quercetin is the aglycone itself, and, before absorption into the enterocyte, any attached chemical groups, such as sugars, must be removed. This is achieved by brush border enzymes, such as lactase phloridzin hydrolase, which remove glucose groups from flavonols. Paradoxically, quercetin glycosides are generally more bioavailable than the aglycone since the latter is more insoluble in the lumen of the gut. Since the brush border enzymes are specific for glucose, quercetin glucosides are absorbed more quickly than other types of glycosides, for example, rutin (quercetin-3-*O*-rutinoside), which can only be deglycosylated to quercetin aglycone by enzymes from the gut microbiota. After absorption by enterocytes, quercetin is glucuronidated, sulfated, and/or methylated. Once absorbed, quercetin enters the bloodstream and appears as various different chemical species, including methylated forms. Quercetin derivatives, such as rutin, which are not absorbed in the small intestine, pass to the colon, where they undergo deglycosylation by α-rhamnosidases and β-glucosidases produced by the gut microbiota. Quercetin bioavailability has inter-individual differences; it is predicted that dietary history, genetic polymorphisms, and variations in gut microbiota metabolism would play significant roles [150].

The therapeutic potential of curcumin is still debated due to a relatively poor bioavailability in humans, even when administered at high dosage (12 g/day) [151]. The poor bioavailability is also exacerbated by the curcumin bindings to enterocyte proteins that can modify its structure [152,153].

The fate of anthocyanins after oral administration follows a unique pattern rather different from those of other flavonoids. Anthocyanins could be absorbed from the stomach, as well as intestines. Some anthocyanins are better absorbed when taken with fruit with a substantial proportion of sugars. Fasting is the area of greatest intestinal absorption of anthocyanins. Anthocyanins appear quickly in plasma, bile, and brain. The metabolites and their activities are partially known. Monoglucuronides are prevalent in the urine, followed by sulphates. Anthocyanins, such as cyanidin-3-glucoside and pelargonidin-3-glucoside, could be absorbed in their intact form into the gastrointestinal wall; undergo extensive first-pass metabolism; and enter the systemic circulation as metabolites. These metabolites could be responsible for the health benefits associated with anthocyanins. Some anthocyanins can reach the large intestine in significant amounts and undergo decomposition catalyzed by microbiota. In turn, these decomposition products may contribute to the health effects associated with anthocyanins in the large intestine [154].

Since polyphenols are extensively modified, and the forms appearing in the blood and tissues are usually different from the forms found in plants; thus, greater attention should be focused on exploring the potential biological activity of polyphenol metabolites, as well as deepening the interaction with the microbiome. In understanding bioavailability of polyphenols, the differences between animal models and human should be taken into account. In order to verify the anti-obesity benefits of polyphenols in in vivo conditions, there is still a lot of scientific research to be done, and well-designed and adequately powered human studies, that undoubtedly verify health-promoting activity of polyphenols in in vivo conditions, are necessary.

## 5. Concluding Remarks

Obesity is a multifactorial disease; there are several predisposing factors, such as genetics, prenatal, and perinatal influences, postnatal weight gain, lack of physical activity, and unhealthy dietary patterns. Consumption of fruit and vegetable has shown promising results in weight management. Fruit and vegetable are polyphenolic-food, they have relatively low energy densities and are high in fiber, and their intake is inversely related to risk of obesity and other chronic diseases [155]. Moreover, polyphenols interact synergistically, not only with other polyphenols, but also with other food components. An example is the traditional Mediterranean diet; this diet is characterized by a high intake of foods rich in polyphenols, including extra virgin olive oil, walnuts, red wine, vegetables, fruit, legumes, and whole grains and adherence to the Mediterranean diet seems to have an anti-obesity effect [156,157,158].

Health effects of polyphenols depend on the quantity consumed and their bioavailability. The bioavailability of polyphenols is very important; after being modified and metabolized by enzymes, their concentration in tissues and biological fluids is quite low. Moreover, there are variations in the individual metabolism of polyphenolic substances; subjects showed either very fast or very slow pharmacokinetics for the same polyphenolic extract, with implications on bioavailability and potential health effects within the body [159]. It is therefore necessary to find a biomarker that accurately reflects the concentration of polyphenols after their absorption and metabolism.

Considering in vitro and in vivo studies, the mechanisms involved in weight loss in which polyphenols could play a role are: activation of β-oxidation processes; induction of satiety; stimulation of energy expenditure; inhibition the differentiation of adipocytes; promotion of apoptosis of adipocytes, increase of lipolysis; and improvement of lipid metabolism disorder (Figure 7). However, the optimal polyphenols dosage capable of maximizing health benefits without raising toxicity issues remains to be elucidated. Generally, evidence for the effects of polyphenols on obesity parameters in humans is still not convincing, possibly due to divergence among study designs, characteristics of the participants, and metabolic pathways. Further randomized controlled trials are needed to confirm the promising protective effects of polyphenols on weight gain and obesity. This research field could be useful for establishing health care and health objectives for overweight, as well as for providing dietary recommendations for individuals.

In conclusion, polyphenolic-food is increasingly looked upon as a valuable alternative or a support for synthetic drugs, as evidenced by a growing number of clinical trials regarding the use of phenolic compounds and polyphenol-rich extracts.

## Figures and Tables

**Figure 1 ijms-21-05642-f001:**
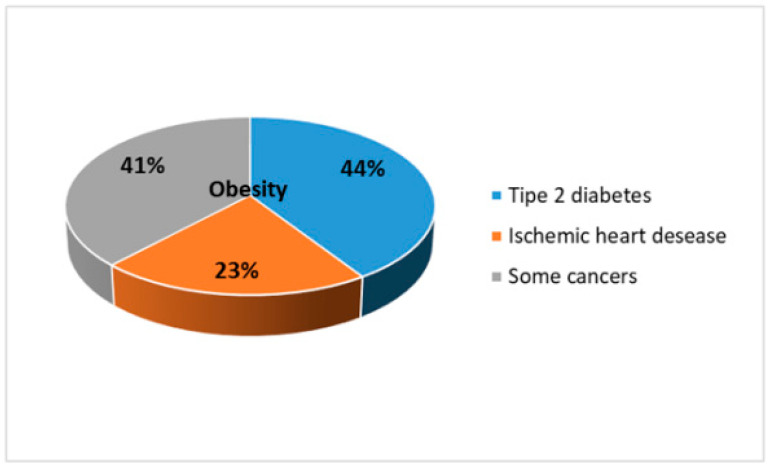
The main pathologies deriving from obesity.

**Figure 2 ijms-21-05642-f002:**
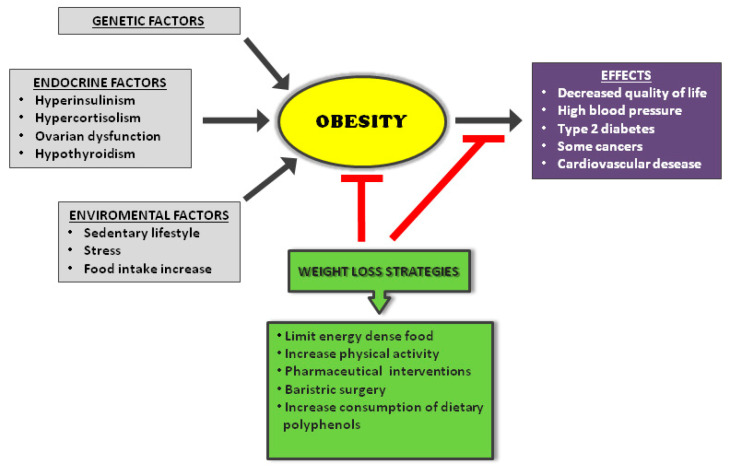
Causes and consequences of obesity with weight loss strategies.

**Figure 3 ijms-21-05642-f003:**
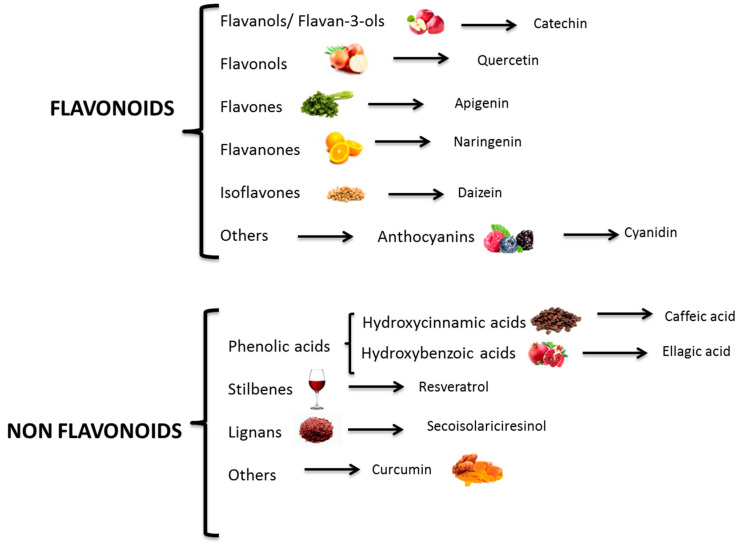
Polyphenols classification.

**Figure 4 ijms-21-05642-f004:**
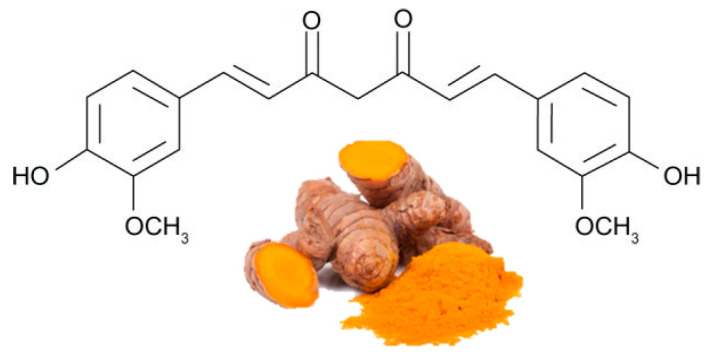
The major sources of curcumin and its chemical structure (keto form).

**Figure 5 ijms-21-05642-f005:**
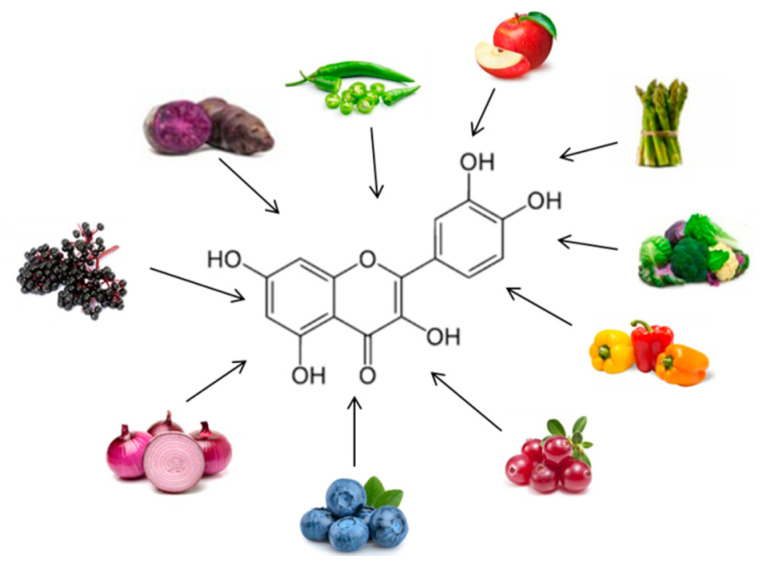
Chemical structure and major sources of quercetin.

**Figure 6 ijms-21-05642-f006:**
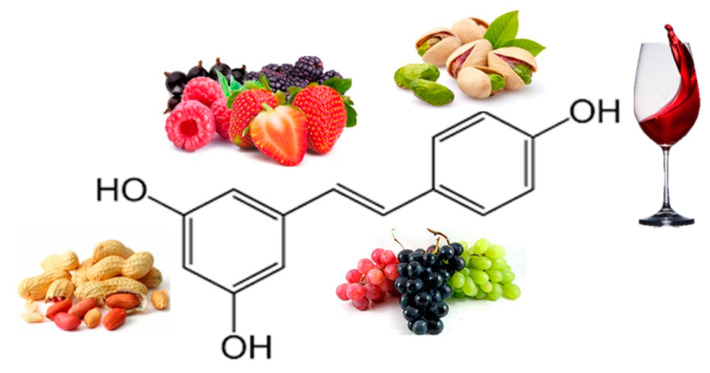
Chemical structure and major sources of resveratrol.

**Figure 7 ijms-21-05642-f007:**
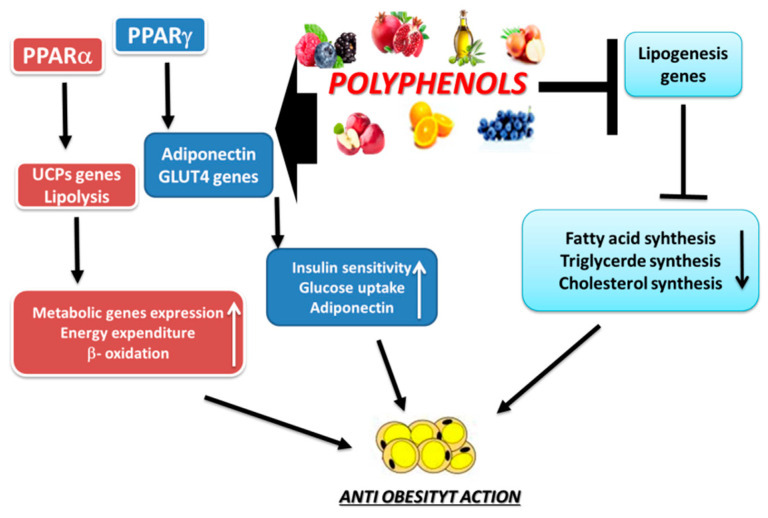
Some anti-obesity effects of polyphenols intake.

**Table 1 ijms-21-05642-t001:** Anti-obesity actions of curcumin, quercetin, and resveratrol.

POLYPHENOL	Human Trials	Effects	References *
**Curcumin**	Randomized, double-blind, placebo-controlled, crossover trial 30 obese individuals 30-day treatment of curcumin (1 g/day)	= Weight, BMI, % of Body fat ↓ TG	65
	Randomized crossover trial 30 obese individuals 30-day treatment of curcumin 1 g/day for 4 weeks	↓ Serum levels of VEGF ↓ IL-1*β* and IL-4	66
	Randomized double-blind placebo-controlled cross-over trial 30 obese individuals Curcumin supplementation (1 g/day for 30 days)	↓ Oxidative stress burden	67
	Randomized, controlled study. 44 overweight subjects 30 days with curcumin	↑ Weight loss ↓↓ % of Body fat ↑ Waistline reduction, ↓↓ Hip circumference ↓↓ BMI	68
	Randomized placebo-controlled clinical trial 60 Overweight and obese female adolescents. 500 mg tablet per day (95% curcumin)	↓ BMI ↓ Waist circumference ↓ Hip circumference ↓ LDL	69
**Quercetin**	Double-blind crossover study overweight-obese subjects 150 mg/d quercetin for 8 weeks	↓ Waist circumference ↓ Postprandial systolic blood pressure ↓ TG	76
	Double-blinded, placebo-controlled cross-over trial 70 Overweight-to-obese subjects 162 mg/d quercetin 6-week treatment periods	↓ Ambulatory blood pressure	77
**Resveratrol**	Randomized double-blind crossover study 11 obese men 150 mg/day resveratrol for 30 days	↓ Insulin, Plasma Fatty Acids, ↓ TG, Glucose, Leptin	85
	11 healthy obese men 30 days (150 mg resveratrol/day)	↓ Adipocyte size	86

* Anti-obesity human studies. IL: InterLeukin; BMI: Body Mass Index; TG: TriGlyceride; LDL: High Density Lipoprotein; VEGF: Vascular Endothelial Growth Factor.

**Table 2 ijms-21-05642-t002:** Anti-obesity actions of some polyphenolic-food.

Polyphenolic-Food	Polyphenols Contained	Effects	Reference *
**Soy beans**	Isoflavones	↓ Serum leptin, TNF-α ↑ Adiponectin	132
		↓ BMI	133
**Black soybean**	Isoflavones Anthocyanins	↓ TNF-α, MCP-1	134
**Red orange**	Anthocyanins Citrus flavonoids	↓ CRP, TNF-α	98, 108
**Mulberries**	Anthocyanins Quercetin Chlorogenic acids	↓ Body weight ↓ Waist circumference ↓ Thigh circumference ↓ Insulin and glucose curves	116
**Strawberry**	Anthocyanins	↓ IL-6 ↓ Oxidized -LDL	118, 119
**Grapefruit**	Citrus flavonoids	↓ Body weight ↓ Waist circumference ↓ Systolic blood pressure	105
**Dried apple**	Flavanols	↓ BMI, TC	122
**Apple**	Flavanols Phenolic acids	= BMI, waist circumferences ↓ Body weight	123
**Green tea**	Flavanols	↓ BMI, waist circumference, glucose ↓ Total cholesterol, LDL-C, TG ↑ Total antioxidant level ↑ Adiponectin	94,98
**Onion**	Quercetin	↓ Body weight ↓ % Body fat ↓ BMI	124

* Anti-obesity human studies. CRP: C-reactive protein; MCP-1: monocyte chemoattractant protein-1; TNF-α: tumor necrosis factor-α; IL: interleukin; BMI: body mass index; TC: total cholesterol; TG: triglyceride; LDL-C: low density lipoprotein-cholesterol.

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
