# Peer review of "Anti-Obesity Effects of Polyphenol Intake: Current Status and Future Possibilities"

_ijms, 2020, doi:10.3390/ijms21165642_

Round 1
Reviewer 1 Report
This review manuscript summarized current anti-obesity effects of polyphenols and provided useful information to the readers. The subject of this review is interesting; the following point should be addressed:
- The content in the “weight loss strategies” seems not to fit well in the review topic. I would suggest to summary this long paragraph into one or two sentences and make the content focusing on the activities of polyphenols on obesity.
Some minor points are
- There are numerous grammatical errors including in the abstract and text of the manuscript.
- The plant names should be italic.
Reviewer 2 Report
Detailed recommendation:
Which title is correct?: Anti-Obesity Effects of Polyphenol Intake: Current Status and Future Possibilities or The Role of Polyphenols in Human Obesity: a Review of Current Evidence??????
Figure 1 should be bigger
Provide more information on the health effects of obesity and its environmental causes
Provide more information about chemical structure and properties of polyphenols.
Table 1 and 2 are unredable
In my opinion images of polyhenols (resveratrol, quercetin ect.) are necessary.
Round 2
Reviewer 2 Report
Manuscript is correct now.
This manuscript is a resubmission of an earlier submission. The following is a list of the peer review reports and author responses from that submission.
Round 1
Reviewer 1 Report
General Comment.
Obesity is a major health problem. There is an urgent need for global action to reduce its prevalence. The authors of the review propose polyphenols as therapeutic agents or supplements to fight obesity (see last sentence of the abstract). However, this idea is not well reflected throughout the article. There is a lack of consistency in the message this review is intending to convey. The authors should make it clear throughout the article that this is the central idea. In my view, this has not been achieved.
Many of the polyphenols have antioxidant properties, but, depending on the amount ingested, they may have a pro-oxidant action. The administration of these compounds as isolated therapeutic agents or as supplements has obvious risks if the pertinent studies are not done. Although the subject of this review is interesting, I do not think it addresses fundamental issues.
Author Response
Dear REVIEWER,
thanks for your time spent reading the manuscript.
In attachment, I send answers to your comments.
Regards,
Stefania D'Angelo

Reviewer 2 Report
In this review manuscript, the authors summarized current anti-obesity effects of certain polyphenols in mainly focusing on human trials. Basically, the manuscript is well written and provides useful information to the readers. However, following points should be addressed to make a better product.
Major point
1. As the authors mentioned, bioavailability of polyphenols is important issue, but information about the bioavailability is lacked in the manuscript body. As the authors know, quercetin, curcumin, and anthocyanins are hardly absorbed into the body per se. They receive metabolic conversion to form conjugates and degradation products. Thus, their active form is unclear. Please add some info about bioavailability of polyphenols in each section.
Minor points
2. Abstract line 13: increased energy expenditure is may be decreased one.
3. Some abbreviations are used without definition; e.g., FTO in line 109, CPT in line 183, and GTE in line 198. Please define these abbreviations.
Author Response
Dear Reviewer,
thanks for your accurate and important comments.
In attachment, I send answers to your comments.
Regards,
Stefania D'Angelo

Reviewer 3 Report
This manuscript provided a timely and comprehensive review of the anti-obesity effects of polyphenol intake. The molecular mechanisms involved in the effects were also reported. The authors interpreted and presented the relevant results correctly and balanced the views of the polyphenol bioactivities, however, there are still a couple of major concerns as follows:
1. It is necessary to differentiate if the anti-obesity effects caused by polyphenol compounds or polyphenol natural products because natural product extracts contain much more components than the identified compounds. However, the different types of polyphenols are mixed together in the manuscript. For example, the “catechins vs. green tea extracts”, the “quercetin vs. onion extract” and some others. It is confusing for readers to follow the correct information.
2. The manuscript reported the useful information about the obesity as a huge public health problem and weight loss strategies in the first two sections, but much of the detailed information is redundant because the manuscript focuses on the polyphenol intake. It would be better to make the information in these two paragraphs concise and clear.
Some minor concerns are
1. There are numberous grammarical errors including in the abstract and text of the manuscript.
2. Many sentences are redundant. For example, most polyphenols demonstrate similar pharmacokinetics. A separate sections named pharmacokinetics might make the paper reader friendly.
3. “In vivo” and “in vitro”should be Italic.
Author Response
Dear Reviewer,
I tried to respond to your comments. I hope that changes made are appreciated.
Regards,
Stefania D'Angelo

Round 2
Reviewer 1 Report
I maintain my earlier opinion.
In lines 453 to 459, the authors talk about Artemisia princes (the correct form is princeps), Physalis alkekengi and Diospyros kaki in section 4.7 "Extra virgin olive oil polyphenols". I do not understand what is their relationship with extra virgin olive oil.
Author Response
Dear Reviewer,
I hope you can change your mind.
Anyway, thanks!
Regards,
Stefania D'Angelo

Reviewer 2 Report
The authors well revised the manuscript, but one point is still remained:
Previous comment #3
FTO is defined in the text, but CPT (line 185) is still used without definition, though the authors defined this word in line 318. Please change it to the first appearance position (line 185).
Author Response
Dear Reviewer,
I tried to respond to your comments. I hope that the changes made are appreciated.
Thanks!
Regards,
Stefania D'Angelo

Reviewer 3 Report
I appreciate all of the corrections. However, I am still not certain about the first major concern. The “4. Anti-obesity properties both single polyphenol and some polyphenolic-extracts” lumps together both the purified compounds and the extracts under a single bullet, but it is not clear enough for readers to follow the main information. I would suggest the authors to re-structure the content within this bullet and make the content reading friendly.
Author Response
Dear Reviewer,
Thank you for your comments.
Thank you for appreciating my corrections. I hope you will also appreciate the latest.
I modified the manuscript following your suggestions.
Regards,
Stefania D'Angelo

Round 3
Reviewer 1 Report
The authors have improved the article with the changes they have made.
Reviewer 3 Report
Thanks for the revision. However, the manuscript is not well organized and it is still not clear enough for readers to follow the main information after a simply separation of the purified polyphenols from the natural product extracts. For example:
- The bioactivities of purified compounds are still mixed with the ones of natural product extracts, such as Line 209-213, Line 231-242 and so on.
- Too many experimental results are listed together but lacked of synthesis or clarification. Not sure what exact information that the authors would like to deliver through these information. such as Line 299-324, the anthocyanins and other polyphenolic extracts are mixed together. Line 364-370, are these plants are polyphenol-rich? What type of polyphenols in these plants?
- A lot of information is still misleading, such as line 333-363, the study on soy protein doesn’t reflect the activities of soy polyphenols, please check the accuracy of the information.
It seems more work need to be done with this manuscript.